# Nasogastric tube in mechanical ventilated patients: ETCO$_2$ and pH measuring to confirm correct placement. A pilot study

Samuele Ceruti[1]*, Simone Dell'Era[2], Francesco Ruggiero[3], Giovanni Bona[4], Andrea Glotta[1], Maira Biggiogero[4], Edoardo Tasciotti[2], Christoph Kronenberg[2], Gianluca Lollo[5], Andrea Saporito[2]

1 Department of Critical Care, Clinica Luganese Moncucco, Lugano, Ticino, Switzerland, 2 Service of Anesthesiology, Ospedale Regionale di Bellinzona e Valli, Bellinzona, Ticino, Switzerland, 3 Department of Internal Medicine, Clinica Luganese Moncucco, Lugano, Ticino, Switzerland, 4 Clinical Research Unit, Clinica Luganese Moncucco, Lugano, Ticino, Switzerland, 5 Department of Gastroenterology and Hepatology, Ospedale Regionale di Bellinzona e Valli, Bellinzona, Ticino, Switzerland

* samuele.ceruti@pm.me

**Data Availability Statement:** All relevant data are within the paper and its Supporting Information files.

## Abstract

### Introduction

Nasogastric tube (NGT) placement is a procedure commonly performed in mechanically ventilated (MV) patients. Chest X-Ray is the diagnostic gold-standard to confirm its correct placement, with the downsides of requiring MV patients' mobilization and of intrinsic actinic risk. Other potential methods to confirm NGT placement have shown lower accuracy compared to chest X-ray; end-tidal CO$_2$ (ETCO$_2$) and pH analysis have already been singularly investigated as an alternative to the gold standard. Aim of this study was to determine threshold values in ETCO$_2$ and pH measurement at which correct NGT positioning can be confirmed with the highest accuracy.

### Materials & methods

This was a prospective, multicenter, observational trial; a continuous cohort of eligible patients was allocated with site into two arms. Patients underwent general anesthesia, oro-tracheal intubation and MV; in the first and second group we respectively assessed the difference between tracheal and esophageal ETCO$_2$ and between esophageal and gastric pH values.

### Results

From November 2020 to March 2021, 85 consecutive patients were enrolled: 40 in the *ETCO$_2$ group* and 45 in the *pH group*. The ETCO$_2$ ROC analysis for predicting NGT tracheal misplacement demonstrated an optimal ETCO$_2$ cutoff value of 25.5 mmHg, with both sensitivity and specificity reaching 1.0 (AUC 1.0, p < 0.001). The pH ROC analysis for predicting NGT correct gastric placement resulted in an optimal pH cutoff value of 4.25, with mild diagnostic accuracy (AUC 0.79, p < 0.001).

**Funding:** The authors received no specific funding for this work.

**Competing interests:** The authors have declared that no competing interests exist.

## Discussion

In patients receiving MV, $ETCO_2$ and pH measurements respectively identified incorrect and correct NGT placement, allowing the identification of threshold values potentially able to improve correct NGT positioning.

## Trial registration

NCT03934515 (www.clinicaltrials.gov).

## Introduction

Nasogastric tube (NGT) placement is commonly performed in the critical setting [1]. The procedure is not however free of complications, which tend to occur especially in patients undergoing mechanical ventilation (MV), when the cough reflex has been abolished [2]. The incidence of complications during NGT positioning is around 4% and is characterized by a high morbidity [3], possibly leading to prolonged hospital stay and higher hospital costs, and increased mortality [4].

Currently, the gold standard to confirm NGT correct positioning is the standard chest X-ray [5], which implies the use of ionizing radiation (4 μSv for radiography) and patients' mobilization, which may lead to accidental extubation and hemodynamic instability. Considering that critical patients may require multiple NGT placements or repositioning, the actinic and mobilization-induced risks are not to be neglected.

Several alternatives to chest X-ray have been investigated, none of them resulting in a high diagnostic accuracy; these include the so-called *bubble technique* [6], frozen NGT [7], gastric auscultation [8, 9], aspiration from the NGT [10, 11], gastric ultrasound [12–14], biochemical markers [15, 16] and the use of magnets [17]. Some pilot studies have shown that measuring end tidal $CO_2$ ($ETCO_2$) with a graphic capnometer could be used to determine whether the NGT tip has been erroneously placed at tracheal level [18–22], others that relative pH levels can distinguish between gastric and esophageal NGT positioning [23, 24]. In those studies, however, threshold values of $ETCO_2$ and pH able to discriminate correct NGT positioning have not been determined, the two measurements (pH and $ETCO_2$) have never been combined in the same study, the global accuracy of each methodology was rather low and/or the sample size was insufficient to obtain a statistical significance. The feasibility to implement the two parameters in a hypothetical device, exploiting a double feedback mechanism to detect correct NGT placement with a high accuracy, is an attractive possibility which justifies further investigations.

Aim of the study is to analyze distributions between tracheal and esophageal $ETCO_2$ values and between gastric and esophageal pH values in two separate arms in patients on MV, in order to identify thresholds values at which the correct positioning of NGT can be confirmed with high accuracy.

## Material & methods

This was a prospective, multicenter, observational trial, conducted over a six-months period in two acute tertiary hospitals. The study has been registered (clinicaltrials.gov, NCT03934515) and approved by the regional Ethical Committee (Comitato Etico Cantonale, Bellinzona, Switzerland, Chairman Prof. Zanini–N. CE3548). The cohort consisted of patients undergoing

general anesthesia and receiving MV; inclusion criteria included patients of both sexes, older than 18 years, fasting for at least six hours, undergoing general anesthesia and MV, for whom the need for an oro- or naso-tracheal tube was decided according to clinical criteria. Exclusion criteria were patient's refusal or inability to give informed consent, pregnancy, known ongoing gastric or esophageal bleeding, coagulation impairment (defined as thrombocytes < 50 G/L, fibrinogen < 1.0 g/L, International Normalized Ratio (INR) > 2.5, activated Prothrombin Time (aPTT) > 70 sec tested at the preoperative assessment), history of traumatic brain injury or polytrauma, esophago-tracheal fistulas, esophageal varices, Ear, Nose and Throat (ENT) malformations and/or tumors, history of radiotherapy for ENT tumors. Patients in whom pH and/or $ETCO_2$ values were not measurable for technical reasons were excluded and considered as drop-outs.

## Allocation

Eligible patients were treated according to allocation within site to one of the two groups: group A patients underwent $ETCO_2$ measurement, group B patients underwent pH measurement. In contrast to protocol's expectations, due to COVID-19 pandemic restriction, Clinica Luganese Moncucco was able to enroll a total of five patients for $ETCO_2$ measurements, while the remaining were enrolled at the Anesthesia Service of Bellinzona and Valli Regional Hospital. As expected by the protocol instead, all patients from group B were enrolled in the Anesthesia Service of "Bellinzona and Valli" Regional Hospital.

**Group A.** After anesthesia induction and at the beginning of MV, a suction probe was inserted in the endotracheal tube and tracheal $ETCO_2$ was measured through a capnometer connected to it; both the probe and the NGT presented the same diameter (12 Fr). After the measurement, tracheal secretions were aspirated as usual. Afterwards, the NGT was positioned using the standard approach, and esophageal $ETCO_2$ was measured through a capnometer attached to it. After the measurement, the capnometer was disconnected, while the NGT was left in place as usual; at this time, no chest-X-ray was routinely performed. $ETCO_2$ values were registered in the data sheet and subsequently transferred into a codified electronic database; for each patient, two sets of values were therefore obtained: tracheal and esophageal $ETCO_2$.

**Group B.** After anesthesia induction and at the beginning of MV, NGT insertion was performed according to local protocols. As the NGT was progressively inserted, pH was measured at two different points located at 25 and 40 cm from the teeth, respectively intended as *esophageal* and *gastric levels*; a chest-X-ray was then performed as standard of care to confirm the sites. The assessment from each of the two levels was performed as following: 10 ml of NaCl 0.9% were first injected and then aspired back in the NGT, and the liquid was then placed on litmus paper for pH assessment. All reported values were then registered and archived as described above. Two sets of values were therefore obtained for each patient: esophageal and gastric pH.

## Outcomes

Primary outcome was to analyze values distributions between tracheal and esophageal $ETCO_2$ measurements (group A) and between gastric and esophageal pH measurements (group B). Secondary outcome was to identify, for each distribution, the threshold value at which correct NGT positioning can be confirmed with the best accuracy.

## Statistical analysis

The power analysis was based on the primary outcomes within each of the two groups. For the $ETCO_2$ group, a tracheal value around 40 mmHg was assumed as normal [25–27], while a

normal esophageal value was considered around 20 mmHg [28]. In order to obtain a significant difference between tracheal and esophageal values, with a power of 90% and a significance level of 0.01 (one-tailed paired t-test), we calculated the need for 35 patients; anticipating a 10% drop-out rate, 40 patients were included in the $ETCO_2$ group. With regard to the pH group, an esophageal value of around 7 was assumed as normal [29], while gastric pH was considered normal when ranging from 1.0 to 2.5 [30]. In order to obtain a significant difference between esophageal and gastric pH values, with a power of 90% and a significance level of 0.01 (one-tailed paired t-test), we calculated the need for 30 patients; similarly, anticipating a 10% of drop-out rate, we included 35 patients for each measurement in the pH group.

We tabulated the distribution of baseline variables across the study's sections, summarizing categorical variables by frequencies and percentage and numerical variables either by mean and standard deviations (±SDs) or by medians and interquartile ranges (IQR). Data distribution was verified using a Kolmogorov- Smirnov test. We executed a paired t-test to compare the two proportions, refusing the null hypothesis of no difference between the two if p-value was ≤ 0.01. In order to identify the threshold value of $ETCO_2$ and pH signaling, respectively, endotracheal and gastric NGT positioning with high accuracy, the area under the receiver operating characteristic (ROC) curve was calculated for both $ETCO_2$ and pH values, calculating the sensitivity, specificity and the likelihood ratios for the optimal cut-off point (CP) of the scale (Youden index and Number Necessary to Diagnose, *J* and *NND* respectively) [31]. Beginning from the ROC curve, a "cumulative distribution analysis" (CDA) was performed [32], to better identify a grey zone defined by the values associated with both sensitivity and specificity of 90% [33]. All hypothesis tests were one-tailed and considered significant if p-value was ≤0.01. Statistical analysis was performed using SPSS.26 (IBM, Chicago, IL, USA) for MacOS.

## Results

From November 2020 to March 2021, 85 consecutive patients were enrolled: 40 in the $ETCO_2$ group and 45 in the pH group; 17 dropouts occurred, due to incomplete information sampling during the procedure (such as the impossibility to measure pH). Sixty-eight patients were therefore included in the analysis, 33 in the $ETCO_2$ group and 35 in the pH group (Fig 1); the mean age was 54 years old (min/max 46–62) and 36 (55%) were men. All demographic data are reported in Table 1.

With regard to the $ETCO_2$ *distribution analysis*, 22 (66%) patients were men, 7 (21%) presented a diagnosis of Chronic Obstructive Pulmonary Disease—COPD (4 patients of second degree, 1 patient of third degree); one (3%) patient presented a previous diagnosis of pulmonary embolism. Five (15%) patients had a history of heart disease (two patients with severity New York Heart Association–NYHA—1, three patients with NYHA 2) (Table 1), all with a cardiac ejection fraction (EF) greater than 50%. Mean tracheal $ETCO_2$ was 40 mmHg (SD 7.14), while mean esophageal $ETCO_2$ resulted 11 mmHg (SD 9.3); a t-test score (Fig 2) confirmed a significant difference (CI 99%, 24–33, $p < 0.001$).

Regarding *pH distribution analysis*, 14 (40%) patients were male, 6 (18%) presented a history of hiatal hernia, and 13 (39%) presented a diagnosis of gastroesophageal reflux disease, with 12 (36%) patients receiving Proton Pump Inhibitors (PPI) therapy at the time of data sampling (Table 1); no patient was on enteral feeding during the analysis. Median gastric pH was 3.1 (1.6–4.95), while median esophageal pH resulted 5.15 (4.52–6.0); a t-test score confirmed a significant difference (CI 99%, 0.9–2.9, $p = 0.004$, Fig 3).

A subgroup analysis involving 20 (62.5%) patients without PPI, showed a median gastric pH of 2.45 (1.05–4.05) and a median esophageal pH of 5.05 (4.52–6.0), with a greater difference of t-test score (CI 99%, 1.2–3.1, $p < 0.001$) compared to all patients (Fig 3). A comparison

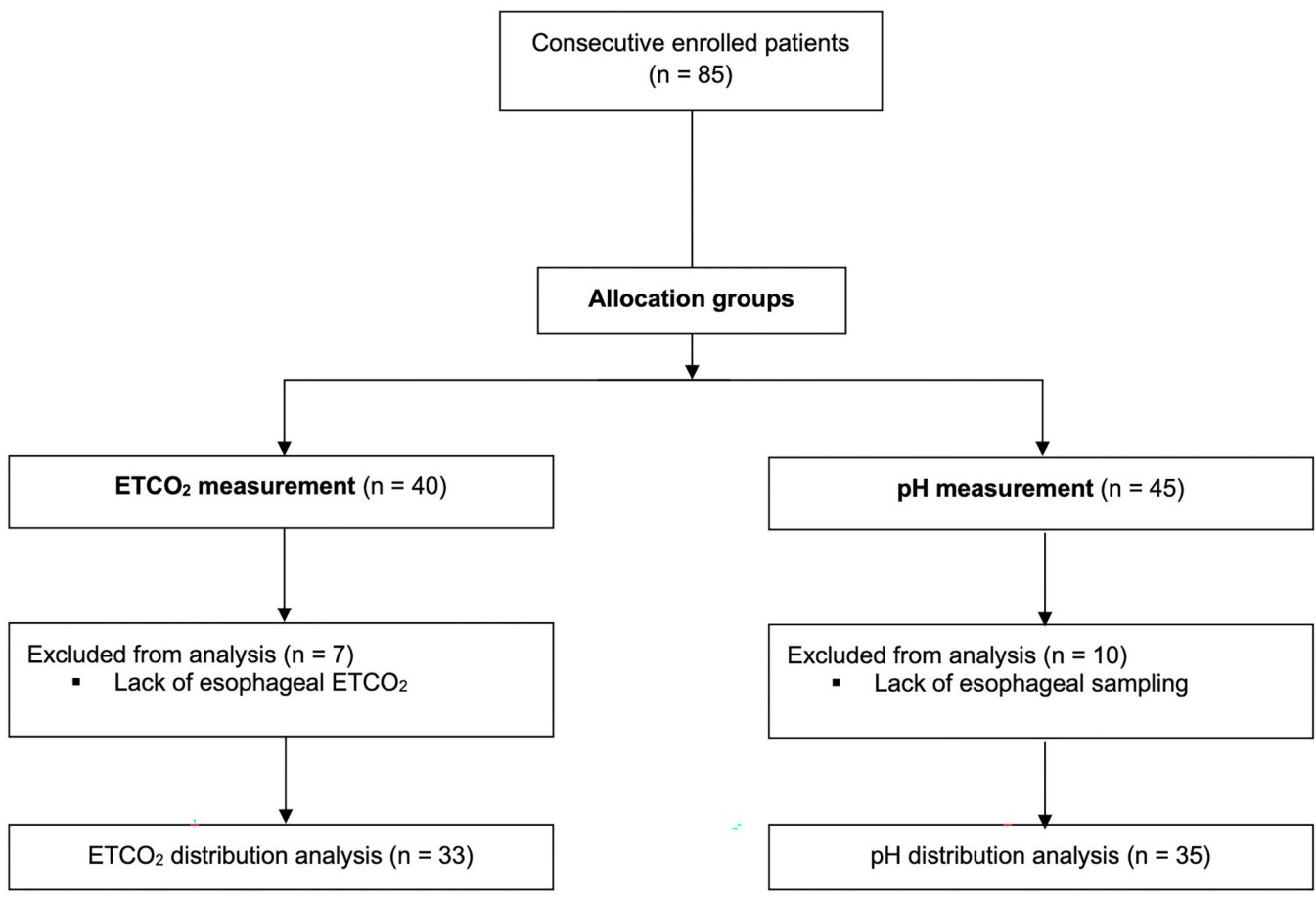

**Fig 1. Patients distribution.** Study patients' allocation and distribution.

between the mean esophageal pH value in all patients and in patients without PPI did not present a significant difference (5.1 vs 4.9, p = 0.265).

## ROC curve analysis

The $ETCO_2$ ROC curve analysis for predicting NGT tracheal misplacement (Fig 4A) demonstrated a perfect diagnostic accuracy with an AUC of 1.0 (CI 95%, 1.0 to 1.0, p < 0.001); the

**Table 1. Demographics characteristic population.**

| | $ETCO_2$ group | pH group | p value |
|---|---|---|---|
| | n = 33 | n = 35 | |
| **Age** | 54 (13.7) | 60 (9.8) | 0.09 |
| **Sex male** | 22 (66%) | 14 (43%) | 0.05 |
| **BMI [Kg/m$^2$]** | 25.4 (6) | 30.6 (7.4) | 0.36 |
| **Systolic arterial pressure [mmHg]** | 145 (29) | 152 (28) | 0.36 |
| **Heart rate [bpm]** | 80 (21) | 77 (17) | 0.43 |
| **Respiratory rate [min]** | 14 (3) | 14 (2) | 0.7 |

Demographic characteristics. Data distribution were expressed as mean ± SD according to Kolmogorov-Smirnov test.

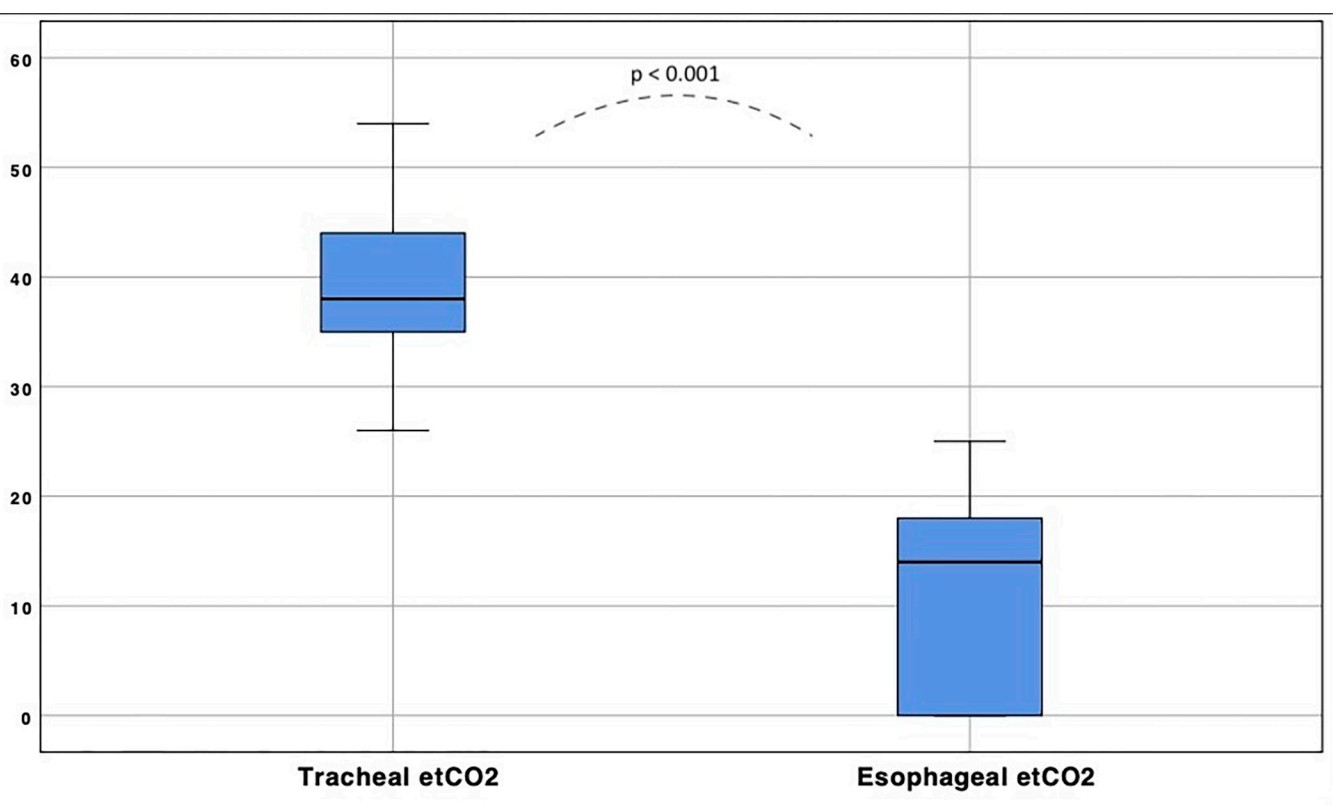

**Fig 2. Tracheal and esophageal ETCO₂ distribution.** Boxplots: The black bar indicates median ETCO$_2$ (38 mmHg and 14 mmHg respectively), while the blue areas include the interquartile ranges for each group.

optimal cutoff value resulted in an ETCO$_2$ value greater than 25.5 mmHg (Youden index J = 1), where both sensitivity and specificity reached 1.0. The pH ROC curve analysis for predicting correct gastric placement (Fig 4B) demonstrated a mild diagnostic accuracy, with an AUC of 0.79 (CI 95%, 0.67 to 0.90, p < 0.001); the optimal cutoff value was a pH below 4.25 (Youden index J = 0.593), with a sensitivity of 0.908 and a specificity of 0.687.

The subgroup analysis involving only patients without PPI confirmed a mild diagnostic accuracy, with an AUC of 0.78 (CI 95%, 0.63–0.93, p = 0.002) and with an optimal cutoff pH value below 3.9 (Youden index J = 0.6). The NND obtained for misplacement of the NGT with the ETCO$_2$ method was 1, while the NND obtained for correct placement of the NGT the pH method was 1.68 (1.66 in patients without PPI).

Grey zone plots were drawn throughout CDA curves starting from the Youden index (Fig 5), between the 90[th] percentages of both sensibility and specificity on the two sigma curves for each ETCO$_2$ and pH; for pH, the gray zone laid between 4.25 and 5.7 (Fig 5), while for ETCO$_2$ no gray zone was identified, as the tracheal and the esophageal distribution did not cross each other (Fig 6).

## Discussion

Nasogastric tube placement in sedated and intubated patients is a procedure potentially associated with dangerous complications. The gold standard to assess correct positioning is Chest X-Ray, which exposes patients to mobilization-related complications, such as devices displacement and hemodynamic and respiratory instability, as well as to actinic risk.

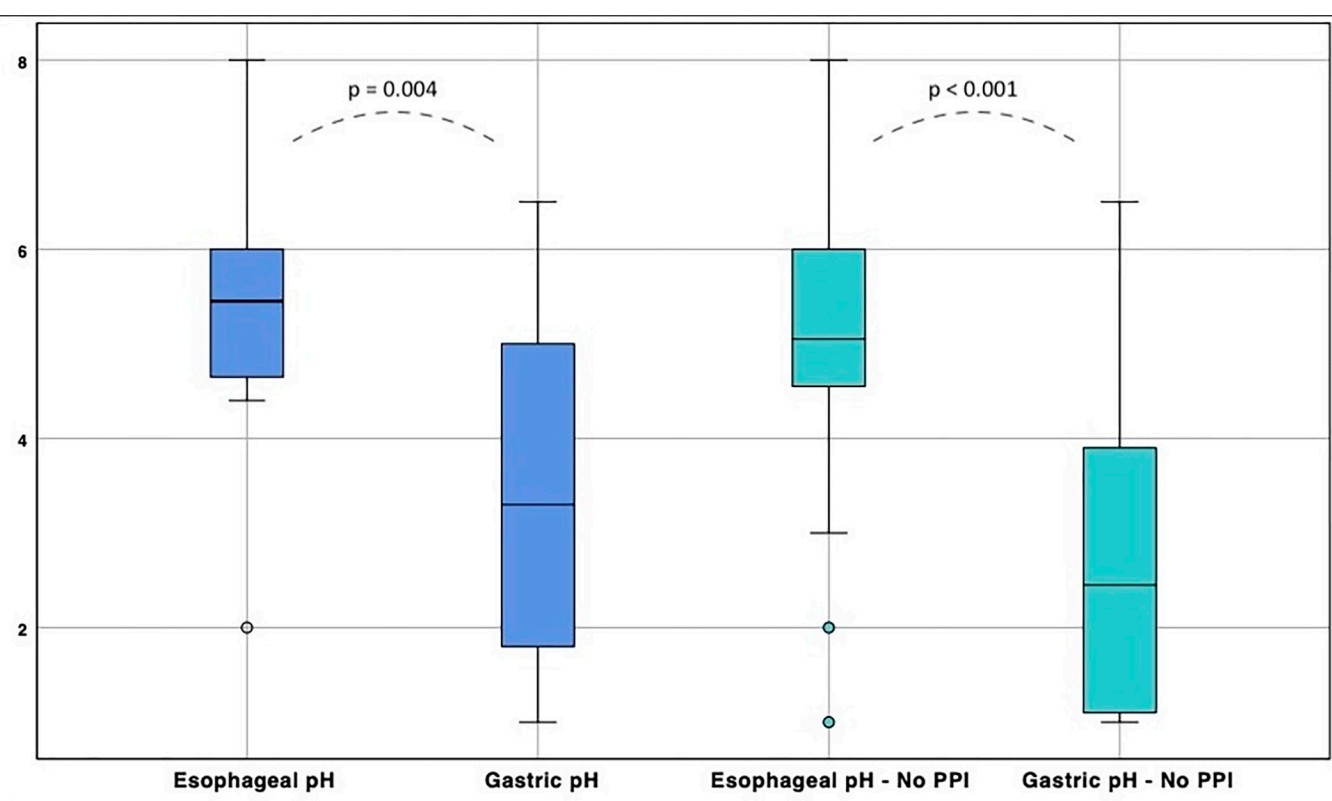

**Fig 3. Measured of esophageal and gastric pH.** Boxplot distribution in all patients and in patients without PPI use. Regarding the whole group analysis, a t-test score confirmed a significant difference between esophageal and gastric values (CI 99%, 0.9–2.9, p = 0.004). The subgroup analysis involving patients without PPI showed a greater difference (p < 0.001) compared to the whole group. The black bar indicates median pH, while the blue areas include the interquartile ranges for each group.

Alternatives to this gold standard, including pH and $ETCO_2$ measurements taken alone, failed to show a superiority in determining correct NGT tip position [18, 23, 34], especially due to the lack of threshold values. Our study analyzed these two techniques, in order to determine if they can accurately detect a correct positioning of the NGT tip. The use of a double feedback mechanism involving both pH and $ETCO_2$ could in fact prove more accurate than just one of the two measurements by itself. In this study, $ETCO_2$ distribution between the trachea and esophagus was evaluated intended as a potential *negative marker* to detect NGT misplacement in the upper airways; at the same time, pH distribution between stomach and esophagus was evaluated as a potential *positive marker* for NGT correct placement. Significant differences between tracheal and esophageal $ETCO_2$ measurements allowed a complete differentiation in the curve plotting distribution. Based on these results, the use of a qualitative capnometer connected to the NGT and set to detect the threshold value of 25.5 mmHg would be a potentially accurate *negative-marker* mechanism for tracheal NGT placement, with a very high sensitivity, thus avoiding any NGT misplacement.

Concerning the differences in results between gastric and esophageal pH, the distributional differences between the two obtained curves is not neat, especially in case of proton pump inhibitors usage, although extremely low pH values were shown to have a high specificity for gastric NGT placement. Fernandez et al published a review of diagnostic studies to test pH of aspirate fluids using a litmus paper; with this method, they evaluated if the NGT had been correctly positioned. It is to be noted that litmus paper color variation could report a value lower

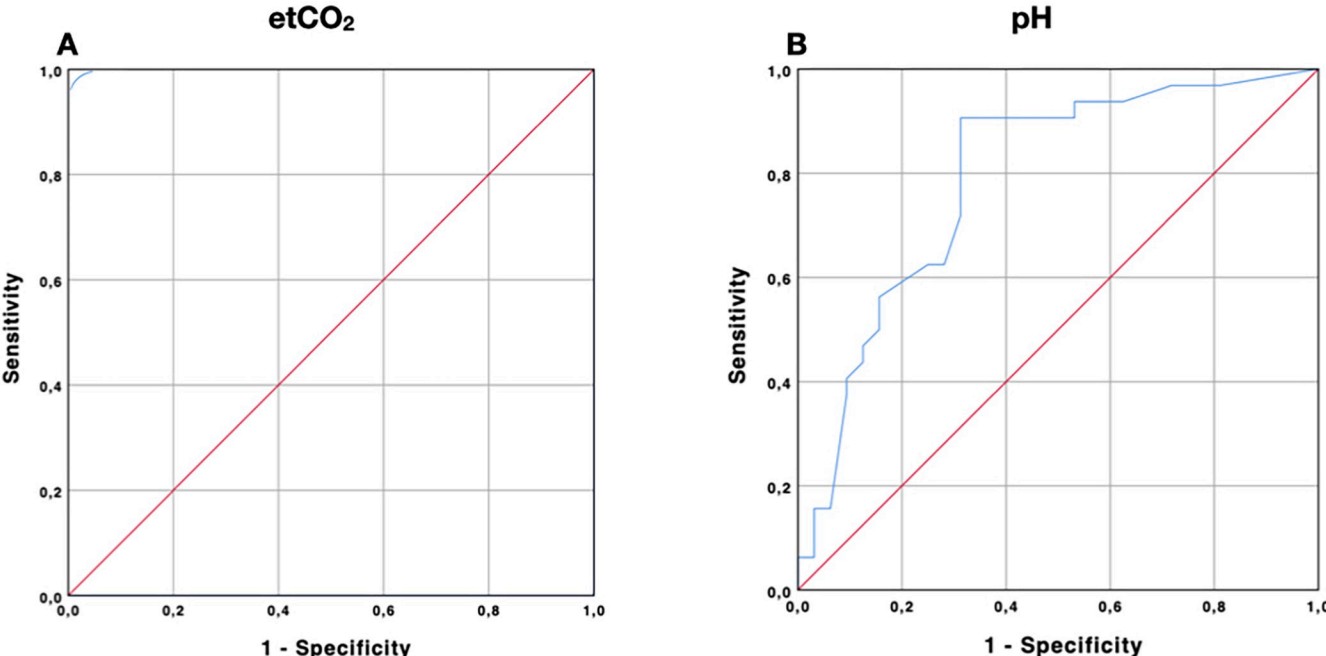

**Fig 4. ROC curves of ETCO₂ and pH.** Receiver operating characteristic (ROC) curves showing the ability of the ETCO$_2$ method (Fig 4A) and pH method (Fig 4B) to respectively identify a tracheal NGT misplacement (ROC AUC 1.0, $p < 0.001$) or a gastric NGT correct placement (ROC AUC 0.79, CI 95% 0.67–0.90, $p < 0.001$).

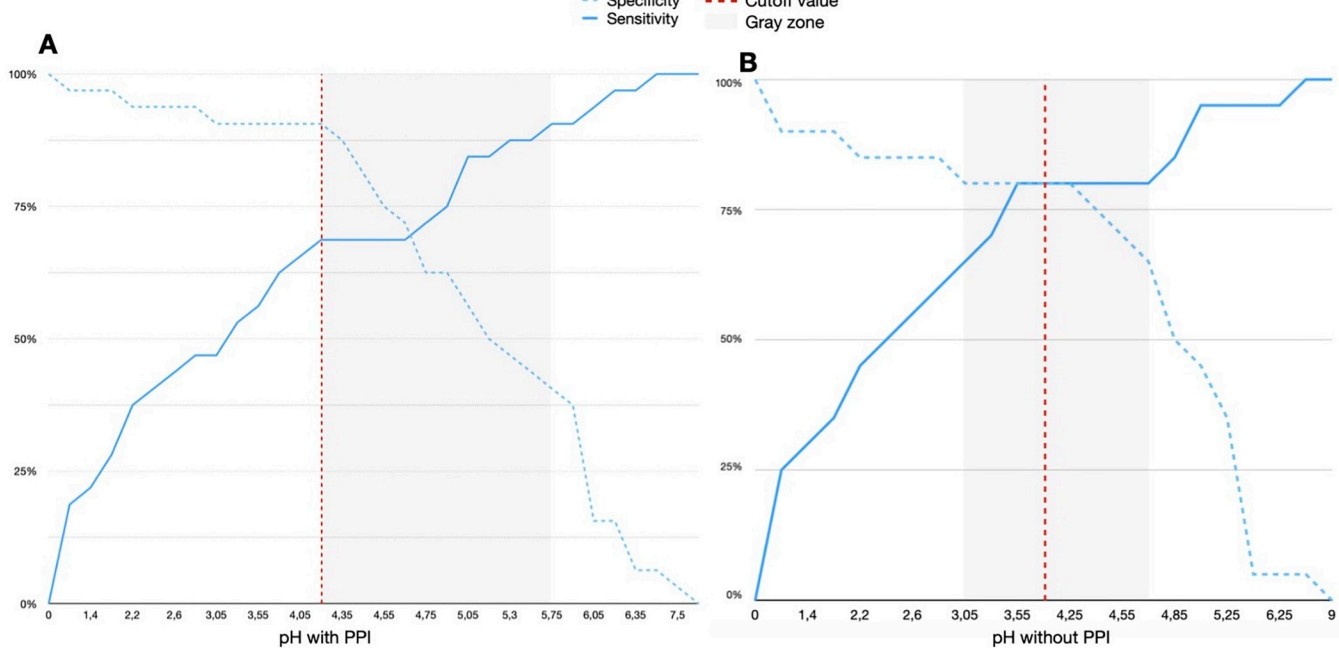

**Fig 5. *Cumulative distribution analysis* of pH detection.** Performed to determine the correct NGT gastric placement with 'Fig 5A' and without 'Fig 5B' PPI use. The red line indicates the cutoff limit according to Youden Index (pH below 4.25 and pH below 3.9, with J = 0.593 and J = 0.6 respectively); the grey zone is shown, with sensibility and specificity of 90%.

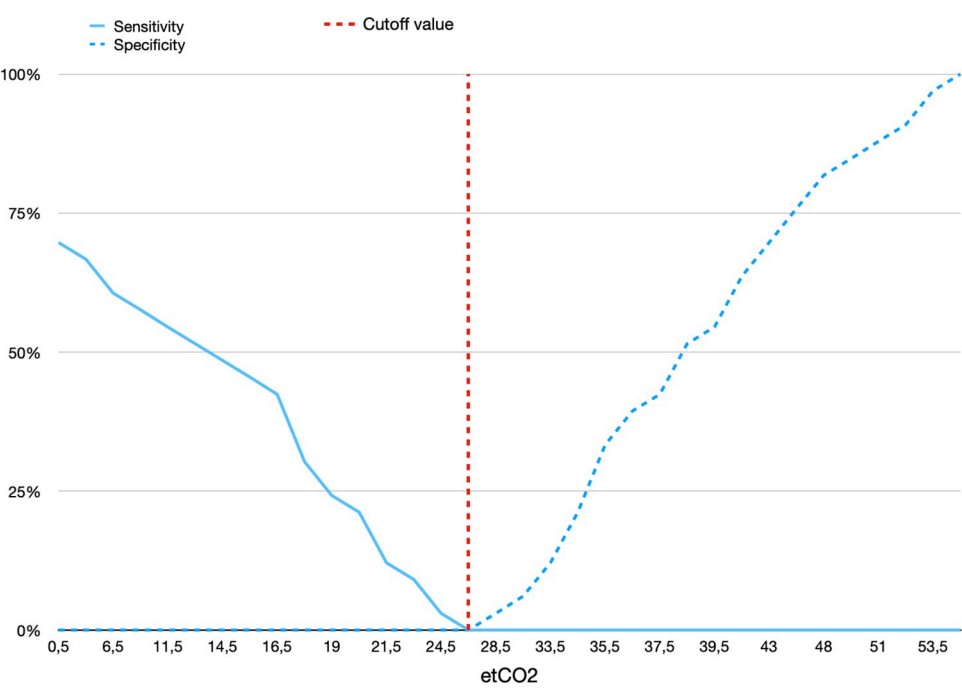

**Fig 6. *Cumulative distribution analysis* of ETCO₂ detection.** Performed to exclude the NGT tracheal misplacement. The red line indicates the cutoff limit according to Youden Index (J = 1).

than the actual gastric pH, due to the paper's limited sensitivity [16]. A recent clinical trial by Gilbertson et al identified the cut off pH of < 5.5 to presume correct NGT positioning in the stomach [23]. Comparing these two studies with our trial, we identified the threshold pH value of 4.25 as the value at which false positive rate is minimized, thus increasing the specificity of this *positive marker*. Noticeably, even if specificity for very low pH appears to be high, the consensually lower sensitivity would affect the global test accuracy, invalidating the *positive marker* mechanism for the detection of correct gastric NGT placement (NND = 1.68), thus leading to potential misses of a correct placement.

Furthermore, analyzing the data based on PPI therapy allows determining an even lower pH threshold for patients not receiving this class of medications (pH of 3.9), nonetheless guaranteeing the same accuracy. In practice, a pH threshold of 4.25 would therefore assure an even better specificity in this subgroup of patients.

Based on our study, a future device capable of combining the presence of a *negative marker* (such as ETCO₂) with a *positive marker* (such as pH) could be accurate enough in identifying the correct positioning of NGTs. Further studies are required to validate the reproducibility of these results with a specific device, whose accuracy also ought to be compared with chest X-ray, the current gold standard.

This study presented some limitations. First, in our trial a small sample of Swiss population was enrolled; for a more robust analysis and validation of the current findings, it could be interesting to perform a larger study involving more hospitals or different geographical areas. Second, this was a preliminary study assessing determined physiological variables; it is still unknown whether a device simultaneously sensing ETCO₂ and pH could determine correct NGT placement with high accuracy. The presumed esophageal and gastric NGTs placement have been determined based on the distance of the NGT tip from the teeth and 2D chest-X-ray; there is not, therefore, complete certainty about NGT tip location; however, NGT placed

at 40 cm from the teeth was conventionally considered into the stomach. Third, in about 20% of patients there was a difficulty concerning the measurements collection, particularly in relation to pH; moreover, due to the small group size it was not possible to perform a sub-analysis concerning the effect of gastric hernia and reflux. In future studies, it will be necessary to implement the usage of the pHmeter, to reduce the rate of dropouts caused by the current limitation of litmus paper. Finally, the accuracy of the pH threshold value for the discrimination between esophageal and gastric NGT positioning resulted suboptimal. The use of a normal saline injection in order to measure pH on the aspirated fluid in case secretions could not be aspirated may have affected pH values in these cases.

## Conclusions

In patients under general anesthesia and receiving MV, $ETCO_2$ and pH measurements to identify NGT tracheal misplacement ($ETCO_2$) and correct gastric NGT placement (pH) allow to identify threshold values potentially able to improve adequate NGT placement detection in MV patients.

## Supporting information

**S1 Appendix. NGT clinical protocol.**
(DOCX)

**S2 Appendix. The preprint version charged on MedRxiv server.**
(DOC)

**S1 Table. NGT codified dataset.**
(XLSX)

## Author Contributions

**Conceptualization:** Samuele Ceruti, Maira Biggiogero, Andrea Saporito.

**Data curation:** Simone Dell'Era, Edoardo Tasciotti, Christoph Kronenberg.

**Formal analysis:** Samuele Ceruti, Francesco Ruggiero, Giovanni Bona, Andrea Glotta.

**Supervision:** Samuele Ceruti, Edoardo Tasciotti, Christoph Kronenberg, Andrea Saporito.

**Writing – original draft:** Samuele Ceruti, Giovanni Bona, Andrea Glotta.

**Writing – review & editing:** Simone Dell'Era, Maira Biggiogero, Edoardo Tasciotti, Christoph Kronenberg, Gianluca Lollo, Andrea Saporito.

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
