## [Decision Letter · Decision Letter 0]

7 Dec 2021

PONE-D-21-20567Nasogastric tube in critical care setting: combining ETCO2 and pH measuring to confirm correct placementPLOS ONE

Dear Dr. Ceruti,

Thank you for submitting your manuscript to PLOS ONE. After careful consideration, we feel that it has merit but does not fully meet PLOS ONE’s publication criteria as it currently stands. Therefore, we invite you to submit a revised version of the manuscript that addresses the points raised during the review process.

<h2>**The study presents the results of the original research; however, the design has some limitations raised by the reviewers. The authors should elaborate on the details of randomization, justify no measurement of CO2 and pH, and non-checking the proper position of NGT with Chest X-ray as the standard of care in all patients. The authors have to justify the validity of measuring oesophagal or gastric CO2. **</h2>

<h2>**The statistics are not described in sufficient detail. The authors should carefully address the concerns about sample size calculation and statistical tests and significance raised by the reviewer. **</h2>

<h2>**The Conclusion is not supported by the data. The conclusion should be derived from the positive findings of the study.**</h2>

The Methods, Results, and manuscript should follow the CONSORT-statement and checklist. 

The authors should remove the redundant text throughout the manuscript. 

 </li> <li> 

The authors should send the next version of the manuscript to a professional Medical English editor who is a native English speaker for substantial English editing and rewriting the manuscript.  

 </li></ul>==============================

We look forward to receiving your revised manuscript.

Kind regards,

Mohamed R. El-Tahan, MD

Academic Editor

PLOS ONE

Additional Editor Comments:

The study presents the results of the original research; however, the design has some limitations raised by the reviewers.

- The authors should elaborate on the details of randomization, justify no measurement of CO2 and pH, and non-checking the proper position of NGT with Chest X-ray as the standard of care in all patients.

- The authors have to justify the validity of measuring oesophagal or gastric CO2.

The statistics are not described in sufficient detail.

- The authors should carefully address the concerns about sample size calculation and statistical tests and significance raised by the reviewer.

The Conclusion is not supported by the data.

- The conclusion should be derived from the positive findings of the study.

The Methods, Results, and manuscript should follow the CONSORT-statement and checklist.

The authors should remove the redundant text throughout the manuscript.

The authors should send the next version of the manuscript to a professional Medical English editor who is a native English speaker for substantial English editing and rewriting the manuscript.

Reviewers' comments:

Reviewer's Responses to Questions

**Comments to the Author**

1. Is the manuscript technically sound, and do the data support the conclusions?

Reviewer #1: Partly

Reviewer #2: Partly

Reviewer #3: Partly

2. Has the statistical analysis been performed appropriately and rigorously? 

Reviewer #1: No

Reviewer #2: I Don't Know

Reviewer #3: I Don't Know

3. Have the authors made all data underlying the findings in their manuscript fully available?

Reviewer #1: Yes

Reviewer #2: Yes

Reviewer #3: No

4. Is the manuscript presented in an intelligible fashion and written in standard English?

Reviewer #1: Yes

Reviewer #2: No

Reviewer #3: No

5. Review Comments to the Author

Reviewer #1: The objective of this prospective, multicenter, observational trial is to determine thresholds in combine measurements of ETCO2 and pH values. The study was approved by the respective IRB/Ethics Committee, and has a valid NCT number (registered within clinicaltrials.org). While the study objectives sound interesting, is important, and on target, a number of shortcomings were observed, in regards to abiding by the CONSORT guidelines for conducting and reporting results of high-quality trials. Some other (statistical) comments were also added.

1. Methods:

Methods reporting require an orderly manner following CONSORT guidelines, without repeating information, such as Trial Design, Participant Eligibility criteria and settings, Interventions, Outcomes, sample size/power considerations, Interim analysis and stopping rules. Randomization (details on random number generation, allocation concealment, implementation), and Blinding considerations should be mentioned explicitly. The authors are advised to create separate subsections for each of the possible topics (whichever necessary), and that way produce a very clear writeup. I see the Authors already made a sincere attempt; however, they are advised to write it carefully, following nice examples in the manuscript below:

https://www.sciencedirect.com/science/article/pii/S0889540619300010

Specific comments below:

(a) For instance, the randomization and allocation concealment should be made very clear (they are NOT the same thing); the trial staff recruiting patients should NOT have the randomization list. Randomization should be prepared by the trial statistician, and he/she would not participate in the recruiting.

(b) More details on randomization needed, like what methods were used. Was it block randomization?

(c) Sample size/Power: The study objectives doesn't compare between the two arms (ETCO2 and pH), but actually compares between the tracheal and esophageal measures for the ETCO2 (and similar groups, gastric and esophageal, for pH). The sample sizes were thus computed "within ETCO2", or "within pH" groups. Also, two sets of values (tracheal & esophageal) were obtained from same patients, for both arms. However, in the sample size/power calculations, authors do not mention the statistical test used, as well as the desired effect size they wanted to power the study upon. I assume, some paired tests used (given that 2 measures were taken for the same subjects)? This needs to be made clear.

It is also not clear if study was powered based on the primary outcomes within both groups.

(d) Statistical Analysis: I do not understand the justification of using a z-test at 1% significance (over a t-test at 5% significance). Also, exactly what test was used (a paired test, or something else) needs to be specified, given that same subjects were used to generate responses, corresponding to tracheal vs esophageal groups.

3. Results:

(a) The authors should check that any statement of significance should be followed by a p-value in the entire Results section. Otherwise, it looks OK.

4. Conclusions and Discussion:

(a) Writeup should reflect that study findings from this trial are based on only a sample (Swiss) population, and allude to future studies enrolling more centers/hospitals, or geographical regions for a more robust analysis, and validation of the current findings.

Reviewer #2: This manuscript describes the results of a prospective multicenter (in fact bi-center) observational trial aiming to determine the accuracy of "EtCO2" and pH measurements during nasogastric tube (NGT) placement in critical care setting to predict right position.

If the research question is appropriate, I have some concerns about the method and the description of results.

First of all, the title includes some wrong informations: "critical care setting" = Most of patients are enrolled in an anesthesia setting; "combining" = the 2 technics are studied separately in 2 distinguished groups. Furthermore, ETCO2 is End tidal CO2 usually used for the CO2 measurement in respiratory airway at the end of expiration and I am not sure it is applicable to esophageal or gastric CO2 measurement.

The authors stated that chest X-rays, the gold standard to confirm correct placement of the NGT, have serious downsides (mobilization and exposure to ionising radiation). This is not completely false but chest X-rays are also useful for many other reasons in ICU and this exam can be done once for all.

The aim to find an alternative technic useful at the bed side and immediately is probably interesting.

In several places in the manuscript, the authors talk about the combine measurement of EtCO2 and pH but it is clear that the patients were randomly assigned either to EtCO2 monitoring in the tracheal tube and in the NGT or to the pH determination in esophageal and gastric position.

The main major concerns about method are:

- How was assessed the correct position in both groups?: no comparison with the gold standard technic is described.

- "Patients in whom pH and/or ETCO2 values were not measurable for technical reasons were excluded and considered as drop-outs": It seems like 17 patients were excluded with reference to figure 1 and the number of analyzed patients. Of note, there is probably a mistake in the line 152 stating "19 dropouts occurred".

The failure rate is high (17 out of 85, 20%) and should be taken in consideration to balance the conclusion. But I should recommend to include these patients in this analysis as a failure of the technic when compared to the gold standard. Here this is a serious biais but not highlighted in the limitation section.

- the correct placement or misplacement of the NGT should be considered by the difference between tracheal and esophageal EtCO2 value or between esophageal and gastric pH values and the respective predictive values should be compared to the chest X-ray as a gold standard. The incidence of tracheal or esophageal misplacement should be given for meaningful interpretation.

Minor comments:

- several abbreviations are not described before first use

- How was the randomization process run out in both centers with allocation of group B patients only in one center?

- The number of patients in each group should be reported in the title column of the table.

The comparison between the groups was not planed in the method and not applicable in most variables. Is it really useful?

- There is some repetitions in results between text and tables or figures. For the same variable, mean or median values are given here and there which is disturbing. Given the small population, it would be preferable to describe the variables in medians with interquartiles

- Discussion, line 229: "Our study combines these two techniques": In this study the 2 technics were studied separately in 2 groups and not combined together

- There is very few comparisons of the results with the literature in the discussion section, maybe due to the lack of data in this field...

- The conclusion is about an hypothesis on the interest of "a device capable of combine the presence of a negative marker to exclude NGT misplacement (like ETCO2) and a positive marker to confirm correct NGT placement (as pH evaluation)". This is not directly issued from the results of the study but more a next step of research.

Reviewer #3: - Relevant topic and objective, worth while studying, good ideas, nice graphics

- But English language is not OK and is compromising the understanding of the article and study by the reader. English language needs to be reviewed both grammatically as with respect of appropriate use in the context of the study. Evident errors or improper word use are highlighted in yellow in an edited PDF version. From time to time I inserted suggestions in textboxes with red characters.

- The study design could have been much better. There are 2 major weaknesses that could easily have been addressed by a more careful study design.

1. why were both measurements (EtCO2 and pH) not performed in all patients and measurements analysed separately; this would have allowed to directly compare both methods.

2. Why was Chest Xray not described and used as a gold standard to be compared with the results of EtCO2 and pH? Are chest X rays available? if sufficiently available why not incorporate in these in an additional analysis and revised article?; I leave it to the editor to decide if such would be advised.

- Abstract reflects insufficiently the study; conclusions in abstract not coherent with methods and study results

- It looks to me as if the use of combined measurement (positive/negative prediction) is a late and awkward interpretation of the results and could have been addressed in a prospective (intentional) way. The advantage of the combined use of both methods is not supported by the way the study is done and described. Furthermore taking the absolute discriminative power of EtCO2 into account the advantage of adding a second method (pH practically inferior) is not clear to me and should be better clarified.

- The EtCO2 method has been recently described in a publication and may therefore have escaped the attention of the writers. This should be referenced and discussed in a revised version.

Therefore I recommend thorough revision of the paper.

See attachments

1- annotated PDF version

2- Word document

6. PLOS authors have the option to publish the peer review history of their article (what does this mean?). If published, this will include your full peer review and any attached files.

Reviewer #1: No

Reviewer #2: No

Reviewer #3: No

---

## [Author Response · Author response to Decision Letter 0]

25 Jan 2022

Dear Dr. Emily Chenette,

On behalf of my co-authors, I have the pleasure to submit the answers to your reviewers’ comments. We appreciated the reviewers’ suggestions that allow us to clarify our key message. Please note that we have uploaded the two versions of the revised manuscript, one in which the new paragraphs were highlighted and an unmarked version, as required. We also uploaded the anonymized data set as Supporting Information file so that the access to our database is provided; all the files meet PLOS ONE’s style requirements. 

ACADEMIC EDITOR: 

• The study presents the results of the original research; however, the design has some limitations raised by the reviewers. The authors should elaborate on the details of randomization, justify no measurement of CO2 and pH, and non-checking the proper position of NGT with Chest X-ray as the standard of care in all patients. The authors have to justify the validity of measuring oesophagal or gastric CO2. 

We thank the editor for this important aspect. As reported by the reviewers, the study was intended as a prospective, multicenter, observational trial, without any randomization. This design was already reported both in the original protocol than on clinicaltrials.gov (NCT03934515). The study was an “allocation within sites” (line 93); the typo regarding randomization that has been entered in the Methods section was removed and modified according to the observational nature of the study (lines 93-98).

• The statistics are not described in sufficient detail. The authors should carefully address the concerns about sample size calculation and statistical tests and significance raised by the reviewer. 

We appreciate the opportunity to clarify this important aspect of the manuscript. The power analysis/sample size was based on the primary outcomes within each group. Due to the design of the study (two sets of values for each patient), a paired t-test was used to make comparisons. According to reviewers’ suggestion, we have deepened and improved the power analysis session, underlining how the study objectives did not compare between the two arms, but actually compared between the tracheal and esophageal measures for the etCO2 and similarly, gastric and esophageal for pH (lines 132-7 and 141-147). Moreover, according to the reviewers’ suggestion, we added the explanation that we utilized one-side paired t-tests, given the nature of the study design, as explained above (lines 127-141). Given the distribution of normal values, and since these are diagnostic tests for which an error of 5% would be too high in relation to a possible position error of the NGT placement, we preferred to run tests and accept them as valid using even more stringent criteria like 1%.

• The Conclusion is not supported by the data. The conclusion should be derived from the positive findings of the study.

We thank the editor for this important aspect. According to reviewers’ suggestion, the Conclusions section in the abstract and in the main text were both revised and made more consistent with the results and discussion.

• The Methods, Results, and manuscript should follow the CONSORT-statement and checklist. 

We thank the editor for this suggestion. As reported by the reviewers, the study was intended as a prospective, multicenter, observational trial, without any randomization; this pilot study was an “allocation within sites” study (lines 93-98), without randomization process; in this scenario, the CONSORT-statement and checklist was not adequate for this kind of trial.

• The authors should remove the redundant text throughout the manuscript. 

We thank the editor for this suggestion; we’ve removed the redundant text throughout the entire manuscript.

• The authors should send the next version of the manuscript to a professional Medical English editor who is a native English speaker for substantial English editing and rewriting the manuscript. 

We thank the editor for this suggestion; we took note of the grammar suggestions and arranged for a further linguistic revision to be carried out by an English native speaker.

Reviewer #1

The objective of this prospective, multicenter, observational trial is to determine thresholds in combine measurements of ETCO2 and pH values. The study was approved by the respective IRB/Ethics Committee, and has a valid NCT number (registered within clinicaltrials.org). While the study objectives sound interesting, is important, and on target, a number of shortcomings were observed, in regards to abiding by the CONSORT guidelines for conducting and reporting results of high-quality trials. Some other (statistical) comments were also added.

1. Methods:

Methods reporting require an orderly manner following CONSORT guidelines, without repeating information, such as Trial Design, Participant Eligibility criteria and settings, Interventions, Outcomes, sample size/power considerations, Interim analysis and stopping rules. Randomization (details on random number generation, allocation concealment, implementation), and Blinding considerations should be mentioned explicitly. The authors are advised to create separate subsections for each of the possible topics (whichever necessary), and that way produce a very clear writeup. I see the Authors already made a sincere attempt; however, they are advised to write it carefully, following nice examples in the manuscript below:

https://www.sciencedirect.com/science/article/pii/S0889540619300010

Response:

We thank the reviewer for this important aspect. As reported by the reviewer, the study was intended as a prospective, multicenter, observational trial, without any randomization. This design was already reported both in the original protocol than on clinicaltrials.gov (NCT03934515). The typo that has been entered in the Methods section was removed and modified according to the observational nature of the study (lines 93-98). 

Specific comments below:

(a) For instance, the randomization and allocation concealment should be made very clear (they are NOT the same thing); the trial staff recruiting patients should NOT have the randomization list. Randomization should be prepared by the trial statistician, and he/she would not participate in the recruiting.

Response:

We thank the reviewer for this important aspect. As reported before, the study was intended as a prospective, multicenter, observational trial, without randomization, as reported both in the original protocol than on clinicaltrials.gov (NCT03934515). The study was an “allocation within sites”; the typo regarding randomization that has been entered in the Methods section was removed and modified according to the observational nature of the study (lines 93-98).

(b) More details on randomization needed, like what methods were used. Was it block randomization?

Response:

We thank the reviewer for this aspect; as discussed before, this was a study with an “allocation within sites” (lines 93-98), without randomization process.

(c) Sample size/Power: The study objectives doesn't compare between the two arms (ETCO2 and pH), but actually compares between the tracheal and esophageal measures for the ETCO2 (and similar groups, gastric and esophageal, for pH). The sample sizes were thus computed "within ETCO2", or "within pH" groups. Also, two sets of values (tracheal & esophageal) were obtained from same patients, for both arms. However, in the sample size/power calculations, authors do not mention the statistical test used, as well as the desired effect size they wanted to power the study upon. I assume, some paired tests used (given that 2 measures were taken for the same subjects)? This needs to be made clear. It is also not clear if study was powered based on the primary outcomes within both groups.

Response:

We appreciate the opportunity to clarify this important aspect of the manuscript. The power analysis/sample size was based on the primary outcomes within each group. Due to the design of the study (two sets of values for each patient), a paired t-test was used to make comparisons. According to reviewer’s suggestion, we have deepened and improved the power analysis session, underlining how the study objectives did not compare between the two arms, but actually compared between the tracheal and esophageal measures for the etCO2 and similarly, gastric and esophageal for pH (lines 127-137 and 143-149).

(d) Statistical Analysis: I do not understand the justification of using a z-test at 1% significance (over a t-test at 5% significance). Also, exactly what test was used (a paired test, or something else) needs to be specified, given that same subjects were used to generate responses, corresponding to tracheal vs esophageal groups.

Response:

We thank the reviewer for this important aspect. According to the reviewer’s suggestion, we added the explanation that we utilized one-side paired t-tests, given the nature of the study design, as explained above (lines 127-137). Given the distribution of normal values, and since these are diagnostic tests for which an error of 5% would be too high in relation to a possible position error of the NGT placement, we preferred to run tests and accept them as valid using even more stringent criteria like 1%.

3. Results:

(a) The authors should check that any statement of significance should be followed by a p-value in the entire Results section. Otherwise, it looks OK.

Response:

We thank the reviewer for this aspect. We have checked and inserted all p-values in the entire Results section.

4. Conclusions and Discussion:

(a) Writeup should reflect that study findings from this trial are based on only a sample (Swiss) population, and allude to future studies enrolling more centers/hospitals, or geographical regions for a more robust analysis, and validation of the current findings.

Response:

We thank the reviewer for this important consideration; we have inserted this indication in the Discussion, in the limit section (lines 268-270).

Reviewer #2

This manuscript describes the results of a prospective multicenter (in fact bi-center) observational trial aiming to determine the accuracy of "EtCO2" and pH measurements during nasogastric tube (NGT) placement in critical care setting to predict right position.

If the research question is appropriate, I have some concerns about the method and the description of results.

First of all, the title includes some wrong informations: "critical care setting" = Most of patients are enrolled in an anesthesia setting; "combining" = the 2 technics are studied separately in 2 distinguished groups. Furthermore, ETCO2 is End tidal CO2 usually used for the CO2 measurement in respiratory airway at the end of expiration and I am not sure it is applicable to esophageal or gastric CO2 measurement.

Response:

We thank the reviewer for these considerations. According to reviewer suggestions, we changed the title removing "critical care" and focusing on the fact that the patients were on mechanical ventilation (MV). Similarly, for what concerns "combining" in the title, we agree with the reviewer and we have removed it from the title. Concerning the consideration of CO2 measurement, we disagree with the reviewer; although ETCO2 is a ventilator measure, it is known that accidental esophageal intubation events are not infrequent (Benumof JL. Interpretation of capnography. AANA J. 1998 Apr;66(2):169-76. PMID: 9801479). In these cases, it is possible to record ETCO2 values, even if very low, as a diagnostic method to confirm an extra-tracheal intubation. Our study has specifically positioned the CO2 detector to the NGT positioned into the esophagus, in order to measure esophageal CO2 values and to be able to determine the difference with the tracheal CO2 values, in order to analyze the CO2 distribution in the trachea and in the esophagus and then determine a "threshold value" that is able to distinguish with high accuracy the NGT positioning in these different two sites.

In several places in the manuscript, the authors talk about the combine measurement of EtCO2 and pH but it is clear that the patients were randomly assigned either to EtCO2 monitoring in the tracheal tube and in the NGT or to the pH determination in esophageal and gastric position.

Response:

We thank the reviewer for this important consideration; to avoid any misunderstanding, we removed the word “combining” throughout the entire manuscript, to eliminate any confounding factor.

The main major concerns about method are:

- How was assessed the correct position in both groups?: no comparison with the gold standard technic is described.

Response:

We appreciate the opportunity to clarify this important aspect of the manuscript. The study method involved open measurements of ETCO2 in the endotracheal tube (trachea site) and in the NGT (esophagus site). Therefore, the different location was chosen by the operator. As for the pH, the use of cm from the dental arch was used for the esophageal (25 cm) and gastric (40 cm) distance, with radiological control as standard. We thank the reviewer for the suggestion and we have implemented this aspect in the text (lines 111-116).

- "Patients in whom pH and/or ETCO2 values were not measurable for technical reasons were excluded and considered as drop-outs": It seems like 17 patients were excluded with reference to figure 1 and the number of analyzed patients. Of note, there is probably a mistake in the line 152 stating "19 dropouts occurred".

The failure rate is high (17 out of 85, 20%) and should be taken in consideration to balance the conclusion. But I should recommend to include these patients in this analysis as a failure of the technic when compared to the gold standard. Here this is a serious biais but not highlighted in the limitation section.

Response:

We appreciate the opportunity to clarify another important aspect of the manuscript. According to the reviewer's suggestions, we included this within the boundaries of the study in the conclusions considerations. Again, we modified the typo of the 19 dropouts patients, who actually were 17. We have also further implemented the limit section of on this aspect (lines 276-278).

- the correct placement or misplacement of the NGT should be considered by the difference between tracheal and esophageal EtCO2 value or between esophageal and gastric pH values and the respective predictive values should be compared to the chest X-ray as a gold standard. The incidence of tracheal or esophageal misplacement should be given for meaningful interpretation.

Response:

We thank the reviewer for this important aspect. We agree with the reviewer, but to date there is no "threshold value" for the etCO2, nor for the pH above/below which it is possible to determine with certainty where the NGT was inserted without a chest-X-ray. In this first study we voluntarily measured in two different sites, with the aim to report the different values of etCO2 and pH and then analyze the distribution of the values and determine the thresholds. In a future study we will analyze if, by applying these thresholds we have determined, we are able to obtain results with high accuracy, compared to gold-standard method.

Minor comments:

- several abbreviations are not described before first use

Response:

We thank the reviewer for this aspect. We have changed the text by inserting the explanation before the first use of the abbreviation.

- How was the randomization process run out in both centers with allocation of group B patients only in one center?

Response:

We thank the reviewer for this important aspect; the study was not a randomized study, as presented in the clinicaltrials.gov and in the protocol. There was a typo in the method, referred to the allocation and not to the randomization. The study was an “allocation within sites” design. 

- The number of patients in each group should be reported in the title column of the table.

Response:

We thank the reviewer for this suggestion, that we have implemented in the table.

The comparison between the groups was not planed in the method and not applicable in most variables. Is it really useful?

- There is some repetitions in results between text and tables or figures. For the same variable, mean or median values are given here and there which is disturbing. Given the small population, it would be preferable to describe the variables in medians with interquartiles

Response:

We thank the reviewer for this aspect. We have simplified the text by removing the redundant parts; furthermore, in this specific context, we have simplified table 1 as suggested by the reviewer.

- Discussion, line 229: "Our study combines these two techniques": In this study the 2 technics were studied separately in 2 groups and not combined together

Response:

We thank the reviewer for this important aspect. In the data analysis, we did not compare the two arms, but we compared the data between the tracheal and esophageal measures for the etCO2 and similarly, gastric and esophageal for pH. According to reviewer’s suggestion, we have removed all sentences referred to “combination”.

- There is very few comparisons of the results with the literature in the discussion section, maybe due to the lack of data in this field...

Response: 

We thank the reviewer for this suggestion; up today, in the literature there is no similar data; this preliminary study was necessary to implement future studies in which we will build a device capable of combining the two aspects of the two tests we have performed here (etCO2 and pH). In order to carry out these future accuracy studies, through comparison with the gold-standard, it was necessary to have reference values which, not being present in the literature, we measured. We thank the reviewer for his suggestion; we have changed the title, suggesting that this was a pilot study, to furtherly avoid any misunderstanding.

- The conclusion is about an hypothesis on the interest of "a device capable of combine the presence of a negative marker to exclude NGT misplacement (like ETCO2) and a positive marker to confirm correct NGT placement (as pH evaluation)". This is not directly issued from the results of the study but more a next step of research.

Response:

We thank the reviewer for this important aspect. According to reviewer suggestion, we have removed all references to the term "combine" and the conclusions were revised and made more consistent with the results and discussion.

Reviewer #3

- Relevant topic and objective, worth while studying, good ideas, nice graphics

- But English language is not OK and is compromising the understanding of the article and study by the reader. English language needs to be reviewed both grammatically as with respect of appropriate use in the context of the study. Evident errors or improper word use are highlighted in yellow in an edited PDF version. From time to time I inserted suggestions in textboxes with red characters.

Response:

We thank the reviewer for this suggestion; we took note of the grammar suggestions and arranged for a further linguistic revision to be carried out by an English native speaker.

- The study design could have been much better. There are 2 major weaknesses that could easily have been addressed by a more careful study design.

1. why were both measurements (EtCO2 and pH) not performed in all patients and measurements analysed separately; this would have allowed to directly compare both methods.

2. Why was Chest Xray not described and used as a gold standard to be compared with the results of EtCO2 and pH? Are chest X rays available? if sufficiently available why not incorporate in these in an additional analysis and revised article?; I leave it to the editor to decide if such would be advised.

Response:

We appreciate the opportunity to clarify this important aspect of the manuscript. On an ethical and organizational level during the pandemic wave, it was easier to organize only one measurement per patient (etCO2 or pH); in order to avoid overlapping or excessive measurements, with the risk of data confusion and excessive investment of time for each patient during the anesthetic routine, it was preferred to act on each individual patient with only one pair of measurements - tracheal and esophageal measurements for the ETCO2 and gastric and esophageal measurements for pH. The comparison between methods was not studied because we developed a preliminary study to try to obtain threshold values for each method rather than a comparison between methods.

For the chest-x-ray (CXR), it was not performed to all patients: as regards the oro-tracheal intubation in the operating room, it was not performed as this procedure is not standard of care. As far as the positioning of the NGT is concerned, we have implemented this aspect by better emphasizing the lack of CXR only in a certain group of patients.

- Abstract reflects insufficiently the study; conclusions in abstract not coherent with methods and study results

Response:

We thank the reviewer for this important aspect; we proceeded to modify the abstract, making the conclusions more consistent with the results of the text.

- It looks to me as if the use of combined measurement (positive/negative prediction) is a late and awkward interpretation of the results and could have been addressed in a prospective (intentional) way. The advantage of the combined use of both methods is not supported by the way the study is done and described. Furthermore taking the absolute discriminative power of EtCO2 into account the advantage of adding a second method (pH practically inferior) is not clear to me and should be better clarified.

Response:

We appreciate the opportunity to clarify this important aspect of the manuscript. Using etCO2 is a great discriminatory method, but it simply allows you to say "you are not in the trachea". It gives no confirmation that it is in the stomach. In this sense, we intended it as a “negative marker”: it allows to identify a mispositioning, but it does not confirm a correct positioning. In this sense it is necessary to identify a method capable of positively confirming the positioning in the stomach, hence the study with the pH in addition to the etCO2.

As for the “awkward results”, the method has been specially designed in this way before starting the study. Since in the literature it is not possible to confirm the positioning of the NGT in the stomach with high accuracy (except with CXR), our hypothesis is that it is necessary to put together several methods. One of these (etCO2) is a "negative" method, ie capable of detecting mispositioning with high accuracy. Another of these (the pH) is intended as a "positive" method, capable of positively confirming the correct positioning of the probe. In this preliminary study we were interested in identifying the values of etCO2 (between trachea / esophagus) and pH (between esophagus / stomach) in order to find diagnostic thresholds.

- The EtCO2 method has been recently described in a publication and may therefore have escaped the attention of the writers. This should be referenced and discussed in a revised version.

Response

We thank the reviewer for these suggestions, even if unfortunately the description of the suggested methodology does not help us in the study we have performed in the determinism of threshold values.

Finally, we would like to thank the reviewers and the editor for these suggestions, because we feel that the manuscript has improved substantially.

Sincerely,

Samuele Ceruti, MD

---

## [Decision Letter · Decision Letter 1]

7 Mar 2022

PONE-D-21-20567R1Nasogastric tube in mechanical ventilated patients: ETCO2 and pH measuring to confirm correct placement. A pilot study.PLOS ONE

Dear Dr. Ceruti,

Thank you for submitting your manuscript to PLOS ONE. After careful consideration, we feel that it has merit but does not fully meet PLOS ONE’s publication criteria as it currently stands. Therefore, we invite you to submit a revised version of the manuscript that addresses the points raised during the review process.

We look forward to receiving your revised manuscript.

Kind regards,

Mohamed R. El-Tahan, MD

Academic Editor

PLOS ONE

Journal Requirements:

Additional Editor Comments:

The authors should address the remaining minor comments for correcting language errors as suggested by the reviewer below.

Reviewers' comments:

Reviewer's Responses to Questions

**Comments to the Author**

1. If the authors have adequately addressed your comments raised in a previous round of review and you feel that this manuscript is now acceptable for publication, you may indicate that here to bypass the “Comments to the Author” section, enter your conflict of interest statement in the “Confidential to Editor” section, and submit your "Accept" recommendation.

Reviewer #1: All comments have been addressed

Reviewer #2: All comments have been addressed

Reviewer #3: All comments have been addressed

2. Is the manuscript technically sound, and do the data support the conclusions?

Reviewer #1: (No Response)

Reviewer #2: Yes

Reviewer #3: Yes

3. Has the statistical analysis been performed appropriately and rigorously? 

Reviewer #1: (No Response)

Reviewer #2: Yes

Reviewer #3: I Don't Know

4. Have the authors made all data underlying the findings in their manuscript fully available?

Reviewer #1: (No Response)

Reviewer #2: Yes

Reviewer #3: Yes

5. Is the manuscript presented in an intelligible fashion and written in standard English?

Reviewer #1: (No Response)

Reviewer #2: Yes

Reviewer #3: No

6. Review Comments to the Author

Reviewer #1: (No Response)

Reviewer #2: Thanks for the clarification of the study and manuscript

Just a comment: in my opinion comparison between the 2 groups seems not useful.

Reviewer #3: - minor correction to improve unambiguous wording (formulating) as described/suggested in detail in attachment

- minor correction in English language as described in attachment

7. PLOS authors have the option to publish the peer review history of their article (what does this mean?). If published, this will include your full peer review and any attached files.

Reviewer #1: No

Reviewer #2: **Yes: **Philippe Gaudard

Reviewer #3: No

---

## [Author Response · Author response to Decision Letter 1]

25 Apr 2022

Reviewer#2

Thanks for the clarification of the study and manuscript

Just a comment: in my opinion comparison between the 2 groups seems not useful.

AND

Reviewer #3

Introduction:

- Lines 67-69. This sentence suggests this will be the objective of the study which – in my opinion – is not the case (as measurements were not combined but used separately in 2 arms

Response:

We thank the reviewers for this relevant aspect, appreciating the opportunity to clarify this key-point of the manuscript. Effectively, a comparative and associative analysis between the two methods has not been done yet; in this first preliminary paper, it was necessary to determine the threshold values of ETCO2 and pH methods, which has still not reported in peer reviewed paper until today. Once obtained this information, a future paper combining and comparing the two methods will be performed. For this reason, as kindly suggested by the reviewer, we removed indication suggesting any “combination” between two methods, as patients were enrolled in 2 separately arms (lines 65-66, line 73).

-Lines Study 71: The latter (measurements were not combined but used separately in 2 arms) should be explicitly mentioned in the aim of the study. 

Response:

We thank the reviewer for this important aspect. According to previous modifications, we have removed all sentences suggesting this “combining aspect” of the paper, implementing in the text that the ETCO2 and pH measurements were 2 complete separate analysis, without any direct comparations (line 73).

Methods

-Line 104 – 105. As opposed to Group B, NGT positioning was not verified (by Chest X ray). This should be mentioned. 

Response:

We thank the reviewer for this suggestion; we have implemented this aspect in the text.

Results

-Line 185-186 (T-test pH) redundant with Line 180-181; to be improved

Response:

We thank the reviewer for this suggestion; we’ve removed the unnecessary text in the figure legend.

-Has the effect of gastric hernia and reflux been examined in this subset of patients. If YES report. If NOT consider this adding in the discussion section on limitations of the study. 

Response:

We appreciate the opportunity to clarify this important aspect of the manuscript; unfortunately, the small group size did not allow us to perform further sub-analysis; for this reason, we have not reported these data. We have reported this aspect in the limit section.

Discussion

-Line 232-233: “…. when applied together”: as this was not the case in this study the authors should be more precise. Furthermore the authors should consider adding an explanation why this was not done (as they explained in their answer to the first review (reviewer #3) 

Response:

We thank the reviewer for this important aspect. Currently, a comparative and associative analysis between the two methods has not been done yet; in this first preliminary paper, it was necessary to determine the threshold values of ETCO2 and pH methods, still not reported in peer reviewed paper until today. Once obtained this information, a future paper combining and comparing the two methods will be performed. Consequently, we have removed any indication or word suggesting any “combination” between two methods.

- Grammar corrections into lines 55, 65-66, 115-116, 185, 200, 211, 211-212, 215, 218, 251, 278

Response:

We thank the reviewer for all these suggestions. All grammar proposals were evaluated and modified according to reviewer’s hints.

Finally, we would like to thank the reviewers and the editor for these suggestions, because we feel that the manuscript has improved substantially.

---

## [Editor Report · Decision Letter 2]

13 May 2022

Nasogastric tube in mechanical ventilated patients: ETCO2 and pH measuring to confirm correct placement. A pilot study.

PONE-D-21-20567R2

Dear Dr. Ceruti,

We’re pleased to inform you that your manuscript has been judged scientifically suitable for publication and will be formally accepted for publication once it meets all outstanding technical requirements.

Kind regards,

Mohamed R. El-Tahan, MD

Academic Editor

PLOS ONE

---

## [Editor Report · Acceptance letter]

24 May 2022

PONE-D-21-20567R2 

Nasogastric tube in mechanical ventilated patients: ETCO^2^ and pH measuring to confirm correct placement. A pilot study. 

Dear Dr. Ceruti:

I'm pleased to inform you that your manuscript has been deemed suitable for publication in PLOS ONE. Congratulations! Your manuscript is now with our production department. 

Kind regards, 

on behalf of

Professor Mohamed R. El-Tahan 

Academic Editor

PLOS ONE